

# The physics of fault friction: Insights from experiments on simulated gouges at low shearing velocities

Berend A. Verberne[1], Martijn P. A. van den Ende[2], Jianye Chen[3,4], André R. Niemeijer[4], Christopher J. Spiers[4]

[1]Geological Survey of Japan, National Institute of Advanced Industrial Science and Technology, 1-1-1 Higashi, Tsukuba, Ibaraki 305-8567, Japan
[2]Université Côte d'Azur, IRD, CNRS, Observatoire de la Côte d'Azur, Géoazur, France
[3]Geoscience and Engineering Department, Delft University of Technology, Stevinweg 1, 2628 CN Delft, The Netherlands
[4]Department of Earth Sciences, Utrecht University, Princetonlaan 4, 3584 CB Utrecht, The Netherlands

*Correspondence to*: B. A. Verberne (bartverberne16@hotmail.com)

**Abstract.** The strength properties of fault rocks at shearing rates spanning the transition from crystal-plastic flow to frictional slip play a central role in determining the distribution of crustal stress, strain and seismicity in tectonically-active regions. We review experimental and microphysical modelling work aimed at elucidating the processes that control the transition from pervasive ductile flow of fault rock to rate-and-state dependent frictional (RSF) slip and to runaway rupture, carried out at Utrecht University in the past two or so decades. We address shear experiments on simulated gouges composed of calcite, halite-phyllosilicate mixtures, and phyllosilicate-quartz mixtures, performed under laboratory conditions spanning the brittle-ductile transition. With increasing shear rate (or decreasing temperature), the results consistently show transitions from 1) stable, velocity(*v*)-strengthening to potentially unstable, *v*-weakening behavior, and 2) back to *v*-strengthening. Sample microstructures show that the first transition, seen at low shear rates and/or high temperatures, represents a switch from pervasive, fully ductile deformation to frictional sliding, involving dilatant granular flow in localized shear bands, where intergranular slip is incompletely accommodated by creep of individual mineral grains. A recent microphysical model, treating fault rock deformation as controlled by a competition between rate-sensitive (diffusional or crystal-plastic) deformation of individual grains and rate-insensitive sliding interactions between grains (granular flow), predicts both transitions well. Unlike classical RSF approaches, this model quantitatively reproduces a wide range of (transient) frictional behaviors using input parameters with direct physical meaning. When implemented in numerical codes for crustal fault-slip, it offers a single, unified framework for understanding slip patch nucleation and growth to critical (seismogenic) dimensions, and for simulating the entire seismic cycle.

## 1 Introduction

Earthquakes are the result of a sudden release of energy during rapid slip (> 1 m/s) along geologic fault zones in the Earth's crust or upper mantle, which generates seismic waves that can be highly destructive at Earth's surface. Throughout history, earthquakes and associated tsunamis have claimed countless lives and caused severe material and economic damage (Guha-



Sapir et al., 2016), with their impact increasing today as urban populations in tectonically active regions continue to increase. It is therefore of utmost importance to improve prognoses on the frequency, location, and magnitude of future seismic events. This demands sophisticated modelling of earthquake nucleation and dynamic rupture propagation, which in turn requires a
fundamental understanding of fault sliding, or more specifically the internal fault rock shearing mechanisms, that are active under in-situ conditions in the Earth.

Tectonically-loaded faults can exhibit aseismic slip transients ("creep") without producing earthquakes, or else sporadic unstable slip, resulting in slow-slip events or catastrophic failure as the case for earthquakes (Scholz et al., 1969; Peng and Gomberg, 2010). Seismic fault motion of this type can occur at the lithosphere-scale such as along subduction zone megathrusts
(Simons et al., 2011; Nishikawa et al., 2019), at the reservoir scale as in the case of human-induced seismicity (Elsworth et al., 2016; Grigoli et al., 2018), but also within mm- to m-scale samples in the laboratory (Passelègue et al., 2013; Yamashita et al., 2015; Ikari, 2019). The fault zones involved typically show a multi-scale, self-affine structure characterized by shear strain localization into narrow, principal slip zones (PSZs) (Tchalenko, 1970; King, 1983; Sibson, 2003), suggesting that the rheology of the comminuted fault rock or "gouge" within PSZs controls macroscopic fault behaviour. From a mechanistic point of view,
improvement of seismic hazard assessments and forecasting requires rationalization of the physics of the earthquake source, as controlled by the material properties of, and deformation processes active within, sheared fault rock.

Laboratory investigations of fault-slip performed under conditions relevant to Earth's upper-crust are key to probing the physics of fault behaviour and seismogenesis. The mechanical data serves as direct input for empirically-based numerical simulations of fault rupture (e.g., Tse and Rice, 1986; Noda and Lapusta, 2013), while post-mortem observations of recovered
deformed specimens can be used to infer the underlying physical processes controlling deformation (e.g., Gu and Wong, 1994; Heilbronner and Keulen, 2006; Peč et al., 2016). In general, we can distinguish two types of laboratory fault-slip experiments. Firstly, low-velocity friction (LVF) tests are used to investigate both stable fault creep and the early (nucleation) stages of earthquake rupture. These LVF experiments are typically conducted at imposed shearing velocities ($v$) of nm to µm or mm per second, under fixed conditions of normal stress ($\sigma_n$) and temperature ($T$). Secondly, high-velocity friction (HVF) tests are used
to investigate dynamic earthquake slip processes that occur during unstable, runaway slip at slip velocities of 1 to 10 m/s (see Heaton, 1990). In HFV tests frictional heating at the slipping fault interface triggers thermally-activated processes such as pore fluid-pressurization, phase changes and melting, which come to dominate the evolution of fault strength (Rice, 2006; Di Toro et al., 2011; Yao et al., 2016). In recent years, technological improvements in both LVF and HVF apparatus, as well as of electron beam and other instruments used to perform post-mortem micro- and nanostructural analyses, have enabled major
advances in the understanding of fault rock material properties and crustal fault rheology (for reviews see De Winter et al., 2009; Niemeijer et al., 2012; Rowe and Griffith, 2015; Chen et al., 2015a).

In this paper we integrate findings from experimental, microstructural, microphysical, and numerical modelling studies of the frictional behaviour of gouge-filled faults in the low velocity or nucleation regime, conducted at Utrecht University (UU) in the past two or so decades. Our aim is to provide a unified view of the physics of fault friction behaviour at low velocities. We
begin with a summary of key concepts and definitions, followed by a summary of the LVF experimental techniques used at





UU. We go on to present key results from experiments on simulated faults composed of halite-phyllosilicate and phyllosilicate-quartz mixtures, and of calcite. Data from these experiments consistently suggest that low velocity frictional deformation of fault gouge is controlled by a competition between rate-sensitive (diffusional or crystal-plastic) deformation and rate-insensitive sliding interactions (dilatant granular flow) —a competition which was already suggested on the basis of theoretical

considerations by Rutter and Mainprice (1979). This forms the foundation for a unified microphysical modelling approach for low velocity sliding and static healing of gouge-filled faults, described in progressive detail by Niemeijer and Spiers (2007) and by Chen and Spiers (2016), for example, and referred to here for convenience as the "Chen-Niemeijer-Spiers" (CNS) model. We outline the principles of this model, and present some applications and implications in reproducing laboratory data and in numerical simulations of earthquake nucleation and the full earthquake cycle.

## 2 Crustal fault strength and fault-slip models


The strength of Earth's crust is classically approximated using a Coulomb-type, brittle/frictional failure law representing the upper part, which abruptly gives way to ductile deformation (here used synonymously with "plastic" deformation to indicate non-dilatant permanent deformation) below ~10 to 20 km depth depending on the geothermal gradient (Fig. 1) (Byerlee 1978; Brace and Kohlstedt, 1980; Kohlstedt et al., 1995). A brittle-to-ductile transition within this depth range is consistent with

geological and seismological observations of a depth interval in the crust where the majority of earthquakes nucleate, known as the "seismogenic zone" (Sibson, 1982, 1983; Meissner and Strehlau, 1982; Scholz, 1988, 2019). In the fully ductile regime, stress buildup and associated rupture nucleation is inhibited by viscous flow in shear zones, which is achieved by solid state diffusive mass transfer and/or dislocation-mediated deformation mechanisms active at the grain-scale (e.g., Karato, 2008). Within the seismogenic zone and shallower it is now widely accepted that faults display so-called 'frictional-viscous'

deformation at steady tectonic slip rates (Fig. 1), characterized by concurrent operation of frictional (cataclastic) processes that depend linearly on effective normal stress, plus viscous deformation processes (typically pressure solution creep) (Wintsch et al., 1995; Holdsworth et al., 2001; Imber et al., 2008; Takeshita and El-Fakharani, 2013). The relation between frictional-viscous deformation in fault zones and seismogenesis, including the competing effects of cataclasis and pressure solution on failure, creep, compaction and healing – and how these control the depth range of the seismogenic zone – remain subjects of

intensive study in fault mechanics and fault geology (e.g., Gratier et al., 2013; He et al., 2013; Fagereng and Den Hartog, 2016; Gao and Wang, 2017; Collettini et al., 2019; Rattez et al., 2019).

In a strictly phenomenological sense, earthquakes are analogous to the recurring frictional instability that is frequently observed in laboratory rock friction experiments known as "stick-slip" (Brace and Byerlee, 1966). Byerlee (1970) proposed that a frictional instability may arise from sudden weakening of the fault interface combined with a sufficiently low shear stiffness

of the surrounding medium (experimental apparatus or host rock). However, this 'slip-weakening' model does not include a mechanism for the intrinsic fault re-strengthening or 'healing' mechanism required to account for long-term, repetitive slip events (Dieterich, 1979a). To capture the time- and sliding velocity-dependent effects of fault friction, in an empirical way,





Dieterich (1979a, b) proposed a rate and "age" dependent friction model that was later recast by Ruina (1983) in terms of the rate-and-state dependent friction equations (RSF), given by

$$\mu = \mu_{ss}^* + a \ln\left(\frac{v}{v^*}\right) + b \ln\left(\frac{v_0 \theta}{d_c}\right) \tag{1}$$

$$\dot{\theta} = 1 - \frac{v\theta}{d_c} \tag{2a}$$

$$\dot{\theta} = -\frac{v\theta}{d_c} \ln\left(\frac{v\theta}{d_c}\right) \tag{2b}$$

where $\mu$ is the coefficient of friction, defined as shear stress over effective normal stress (ignoring cohesion), $\mu_{ss}^*$ is the steady-state coefficient of friction at a reference sliding velocity $v^*$, $v$ is the instantaneous slip velocity, $a$ is a parameter that quantifies

the "direct effect", $b$ is the parameter that describes the "evolution effect", and $d_c$ is a characteristic or critical slip distance over which the state variable, $\theta$, evolves (Marone, 1998; Scholz, 1998). The state variable $\theta$ has units of time and is thought to represent the average lifetime of grain-scale asperity contacts (at steady-state; Scholz, 2019). The conceptual interpretation of RSF finds its origin in two observations, namely that the true area of contact of any interface is always smaller than the nominal contact area, and that this area of contact changes with time (Dieterich and Kilgore, 1994). The equations for the state

variable time derivative ($\dot{\theta}$; eqs. 2a and 2b) embody two views of how a population of contacts may evolve during slip. Equation 2a is sometimes called the "slowness" or "aging" (Dieterich) law, because in this formulation, the frictional contact area continues to evolve in the absence of slip, whereas in equation 2b, slip is needed for the state variable to evolve. Accordingly, the latter equation is called the "slip" (Ruina) law.

In the case of steady state sliding ($\mu = \mu_{ss}$; $\dot{\theta} = 0$) both the slowness and slip laws reduce to

$$\mu_{ss} = \mu_{ss}^* + (a - b) \ln\left(\frac{v}{v^*}\right) \tag{3}$$

where *(a-b)* represents a dimensionless quantity that characterizes the velocity(*v*)-dependence of the sliding surface. Ruina (1983) showed that for an instability to nucleate spontaneously and repeatedly (i.e., the case of stick-slip), the sliding surface must decrease in strength with increasing velocity, hence be *v*-weakening, characterized by *(a-b)* < 0. In the opposite case *v*-strengthening occurs, characterized by *(a-b)* > 0, which leads to a state of stable sliding (Ruina, 1983; Rice and Ruina, 1983).

Linear stability analysis of a single-degree-of-freedom spring-slider model demonstrated a critical stiffness $K_c$ below which sliding is unstable (Ruina, 1983)

$$K_c = \frac{(b-a) \cdot \sigma_n^{eff}}{d_c} \tag{4}$$

Thus, an instability may occur when the stiffness of the deforming medium falls below $K_c$, and *(a-b)* < 0. Importantly, when applied to natural fault deformation, the *(a-b)*-value or *v*-dependence of the sliding medium is a material property of deforming

fault rock in the fault core, which is strongly affected by coupled thermo-hydro-mechanical-chemical processes in the general





sense. With reference to Earth's crust and Figure 1, the seismogenic zone is believed to represent the depth interval where shear deformation of fault rock leads to *v*-weakening behavior, as opposed to *v*-strengthening above and below.

The RSF approach has enabled a simple and highly successful description of laboratory rock friction behavior over a wide range of conditions (for reviews see Marone, 1998; Scholz, 1998, 2019), and is widely used in numerical simulations of the

earthquake cycle (e.g., Dieterich, 1994; Lapusta and Rice, 2003; Ampuero and Rubin, 2008; Matsuzawa et al., 2010; Noda and Lapusta, 2013; Ohtani et al., 2014). However, laboratory observations and microphysical modelling show that the empirically fitted parameters appearing in the RSF equations, notably the values of *(a-b)* and $d_c$, are not fundamental, independently measurable material constants (Ikari et al., 2016; Aharonov and Scholz, 2018). The implication is that extrapolation to natural conditions not attainable in the laboratory presents a significant source of uncertainty in numerical

modelling (for a discussion see Ide, 2014; Van den Ende et al., 2018). To address this, a (micro)physically based interpretation and description of the processes controlling fault deformation is needed. An example of a mechanistically-based, and microstructurally founded model developed to explain *v*-dependence effects of fault rock friction, proposed by Bos, Niemeijer, and Spiers (Bos and 2002a; Niemeijer and Spiers, 2006, 2007), is based on the accommodation of shear deformation by a combination of frictional and viscous processes. It was demonstrated that if the rates of intergranular compaction ($\dot{\varepsilon}_{cp}$) and

dilatation ($\dot{\varepsilon}_{dil}$) are of comparable magnitude, or $|\dot{\varepsilon}_{dil}| \approx |\dot{\varepsilon}_{cp}|$, this leads to *v*-weakening behavior, whereas under conditions where either process dominates, stable *v*-strengthening occurs. Viewed in this framework, the seismogenic zone corresponds to a depth interval where shear deformation of gouge-filled faults is characterized by $|\dot{\varepsilon}_{dil}| \approx |\dot{\varepsilon}_{cp}|$ (Fig. 1).

**3 Low-velocity friction (LVF) testing methods**

Research on low-velocity rock friction at UU involved fault-slip experiments carried out under pressure-temperatures that

range from ambient surface conditions to those relevant throughout the upper ~20-30 km of the Earth's crust. Because of their critical role in this research, below we summarize the ring- and direct-shear testing methods installed at UU. For details on experimental procedures and data analysis methods we refer to the papers cited below.

Ring shear LVF experiments are conducted using two distinct set-ups that are interchangeable within a single, rotary-shear deformation apparatus, which consists of an Instron loading frame with electrically actuated ram for axial loading (application

of normal stress) plus a rotation drive for imposing shear displacement onto the sample (Figs. 2a, b). The earliest ring-shear assembly was developed in the late 1990's (Bos et al., 2000a), and enabled simulated fault sliding tests at room temperature and at elevated normal stresses and pore fluid pressures (up to ~10 MPa), achieving in principle unlimited rotational displacements. The simulated fault sliding rates that can be achieved depend on the arrangement of gear boxes used, and ranges from 3 nm/s to up to 1 cm/s. The assembly consists of two grooved piston rings (inner diameter 80 mm, outer diameter 100

mm) that grip a ~1-2 mm thick, annular sample layer upon the application of normal stress, with radial confinement facilitated by tightly-fitting inner and outer rings (Figs. 2a, d). This room temperature ring-shear set-up has played an important role in investigations of shear deformation of monomineralic halite and halite-phyllosilicate mixtures (e.g., Takahashi et al., 2017;



Van den Ende and Niemeijer, 2019), and of granular system dynamics using synthetic polymer and glass beads (Kumar et al., 2020).

A later, 'hydrothermal', ring-shear assembly was designed and commissioned in 2002-2005 (Fig. 2b, e), with the aim of enabling high shear strain, rotary-shear tests under pressures, temperatures, and displacement rates representative for the seismogenic reaches of crustal scale faults and subduction megathrusts (Niemeijer et al., 2008; Van Diggelen et al., 2010; Den Hartog et al., 2012a, b). In this setup, the piston-sample assembly is located in a pressure vessel with internal furnace, which in turn is emplaced within the Instron frame with rotation drive (Fig. 2b). A ~1 mm thick sample layer is sandwiched between
a set of grooved pistons, and prevented from extruding by an inner confining ring with a diameter of 22 mm, and an outer confining ring with a diameter of 28 mm (Fig. 2e). The vessel is pressurized with water, which has direct access to the sample thus providing the pore fluid pressure. Experiments in this setup can be conducted at effective normal (axial) stresses up to 300 MPa (provided by the Instron frame), temperatures up to 700°C, and pore fluid pressures up to 300 MPa. The rotary drive system provides simulated fault zone displacement rates ranging from around 1 nm/s to several mm/s. The maximum rotation
or shear displacement that can be achieved is limited by the water cooling and pore fluid systems, but is in practice large (> 100 mm). The hydrothermal ring shear machine has been used extensively in investigations of the shear behavior of rock compositions believed to be widespread along subduction megathrust faults (e.g., Hirauchi et al., 2013; Ikari et al., 2013; Sawai et al., 2016, 2017; Kurzawski et al., 2016, 2018; Boulton et al., 2019; Okamoto et al., 2019, 2020) and in the upper and middle continental crust (Niemeijer and Collettini, 2014; Niemeijer and Vissers, 2014; Niemeijer et al., 2016; Hellebrekers et al.,
175    2019).

Direct-shear tests are carried out using a 'conventional', externally or internally heated, oil-medium, triaxial deformation apparatus, such as that shown in **Figure 2c**. Following the design of Logan et al. (1992), the direct-shear or "69" assembly comprises two L-shaped pistons in a jacketed face-to-face ( ⌞⌐ ) arrangement that sandwiches a cuboid sample (**Figs. 2f**). A soft, near-Newtonian viscous material (such as silicone putty) or soft elastomer fills the voids at the top and bottom of the
assembly (Samuelson and Spiers, 2012; Sánchez-Roa et al., 2016). In the set-up used at UU, the sample measures 35 mm wide by 49 mm long with a thickness of typically ~1 mm. Direct-shear experiments can be carried out at confining pressures (= normal stress) and pore pressures up to 100 MPa, temperatures up to 150°C, reaching shear displacements up to ~6 mm. The direct-shear assembly has proven especially useful for tests employing corrosive pore fluid compositions such as reservoir brine or $CO_2$ (e.g., Pluymakers et al., 2014; Pluymakers and Niemeijer, 2015; Bakker et al., 2016; Hunfeld et al., 2017, 2019).

**4 Low velocity friction experiments on simulated gouges – some case studies**

**4.1 Halite-phyllosilicate mixtures**

Bos et al. (2000a, b) and Bos and Spiers (2000, 2001, 2002a, b) employed the room temperature ring-shear assembly (Fig. 2a, d) to investigate the shear behaviour of brine-saturated, simulated fault gouge composed of (mixtures of) halite and kaolinite. Bos and co-workers' experiments were carried out under conditions favouring rapid pressure solution in the halite-brine





system, which is well-constrained from compaction tests and microphysical modelling (Spiers et al., 1990; Spiers and Schutjens, 1990). Kaolinite was added in varying proportions to investigate the effect on shear behaviour, while simulating the presence of phyllosilicates that are observed to be widespread in natural fault zones (e.g., Wintsch et al., 1995; Holdsworth, 2004). The aim was to elucidate the combined role of pressure solution creep and foliation development in controlling the strength of faults in the upper-crust, viewing the halite-kaolinite mixtures used as a mid-crustal rock analogue (cf. Shimamoto,

1986; Hiraga and Shimamoto, 1987; Chester and Logan, 1990). Velocity (v-) and normal stress ($\sigma_n$-) stepping experiments (Fig. 3a), as well as post-mortem microstructural analyses which revealed a classical, foliated mylonitic (i.e., phyllonitic) microstructure (Fig. 3b), pointed to frictional-viscous flow in the case of halite-kaolinite mixtures, but to purely frictional behaviour in the case of the monomineralic end-member gouges (Fig. 3a). Based on these results, Bos and Spiers (2001, 2002a) proposed a micromechanical model for the combined effect of frictional sliding on phyllosilicate folia, pressure solution of

halite clasts, and dilatation on the foliation (Fig. 3c), which offered the first microstructurally-based interpretation for *v*-strengthening, frictional-viscous flow of gouge-filled faults.

Niemeijer and Spiers (2005) refined the Bos and Spiers' model by incorporating effects of plasticity of phyllosilicate folia and a distributed grain size. Moreover, their experiments used muscovite instead of kaolinite and covered a wider range of sliding velocities. This allowed them to identify a *v*-weakening regime beyond $v = 1$ µm/s (Fig. 3d), characterized by a strong increase

in porosity with increasing *v* (Niemeijer and Spiers, 2005, 2006). Compared with *v*-strengthening samples, those deformed under *v*-weakening conditions showed a chaotic, cataclastic microstructure (Fig. 3e). On this basis it was hypothesized that *v*-weakening results from a competition between dilatation by granular flow and intergranular compaction by pressure solution (Niemeijer and Spiers, 2006). This microphysical model concept was further developed and quantified to enable calculation of steady-state shear strength in the *v*-weakening regime based on physically meaningful input parameters such as the kinetics

parameters for pressure solution, porosity, and dilation angle for granular flow (Niemeijer and Spiers, 2007) (Fig. 3f). By combining the model for frictional-viscous flow with that for *v*-weakening frictional sliding, the lab-observed transition from *v*-strengthening to -weakening with increasing shear displacement rate in halite-phyllosilicate mixtures could be accurately reproduced.

**4.2 Phyllosilicate-quartz mixtures**

The experiments using halite-phyllosilicate mixtures as a fault rock analogue system trigger the inevitable question of whether the same processes and mechanical behaviour really occur within crustal faults. Specifically, the microphysical models developed for *v*-dependence of gouge-filled faults required testing for real crustal fault rock types, under conditions relevant for the seismogenic zone. With this in mind, Den Hartog et al. (2012a, b, 2013, 2014) investigated the frictional behaviour of phyllosilicate-quartz gouge mixtures using the hydrothermal ring shear apparatus (Fig. 2b, d), under P-T conditions broadly

representative for the seismogenic reaches of a subduction zone megathrust. The samples consisted mainly of 65:35 illite:quartz gouge mixtures, but muscovite-quartz gouge mixtures and clay-rich samples derived from the Nankai Oceanic Drilling Project (Leg 190) were also tested. Experiments were carried out at effective normal stresses ($\sigma_n^{eff}$) ranging from 25





to 200 MPa, at pore fluid pressures ($P_f$) of 0 (dry) to 200 MPa, and temperatures ($T$) of 100°C to 600°C. The data consistently showed $v$-strengthening behaviour at relatively low temperatures (up to ~250-350 °C, Regime 1), $v$-weakening at intermediate

temperatures (~250-500 °C, Regime 2), and again $v$-strengthening at the highest temperatures investigated (>500 °C, Regime 3) (Fig. 4a). Such 'three-regime' $v$-dependence with increasing temperature has been observed for granite (Blanpied et al., 1991, 1995) and gabbro (He et al., 2007) gouges, but Den Hartog and co-workers were the first to report this for a realistic megathrust fault rock composition. Moreover, the temperature range in which $v$-weakening was reported is broadly consistent with the temperature-depth range of the subduction seismogenic zone such as in Nankai (Hyndman et al., 1997; Yoshioka et

al., 2013; Okamoto et al., 2019).

The above observations on illite- and muscovite-quartz gouges were explained first qualitatively and later using a quantitative model based on the Niemeijer and Spiers (2007) approach, but employing a phyllosilicate-dominated model microstructure (Fig. 4b) (Den Hartog and Spiers, 2013, 2014; see also Noda, 2016). The change in the sign of $(a–b)$ with increasing temperature was proposed to occur due to changes in the relative importance of thermally-activated deformation of the quartz

clast phase (by stress corrosion cracking and/or pressure solution) versus athermal granular flow of the mixture accompanied by dilatation (Fig. 4b). Moreover, on the basis of widespread experimental observations (Ikari et al., 2011), expected $v$-strengthening effects of frictional slip within the phyllosilicate matrix and foliation were taken into account. Using pressure solution as the controlling thermally-activated process, the experimentally observed 'three-regime' $v$-dependence could be reproduced. In addition, Niemeijer (2018) recently showed a good match between data from constant v shear experiments

using 80:20 quartz:muscovite gouges, and predictions of the "Bos-Spiers-Niemeijer" frictional-viscous flow model (Bos and Spiers, 2002a; Niemeijer and Spiers, 2005) (Fig. 4c). Regardless of the details of the model used, the results of Den Hartog and co-workers, and those of Niemeijer (2018), imply that shear strain accommodation involving a competition between rate-sensitive (thermally-activated creep of clast phases) and rate-insensitive processes (dilatant intergranular sliding), plays a key role in controlling $v$-dependent frictional and frictional-viscous flow of phyllosilicate-quartz mixtures.

**4.3 Calcite**

Motivated by the frequency of destructive earthquakes in tectonically-active carbonate-bearing terranes such as the Apennines (Italy) and the Longmen Shan (China), Verberne et al. (2013, 2014a, b) and Chen et al. (2015b, c, 2020a) investigated the frictional behaviour of simulated calcite(-rich) fault gouge. Initial experiments employing the direct-shear assembly (Fig. 2c, f) were conducted at $T$ = 20-150 °C, $\sigma_n^{eff}$ = 50 MPa, and a pore water pressure $P_f$ = 10 MPa or else under room-dry conditions.

Dry and wet velocity-stepping ($v$ = 0.1, 1, 10 µm/s) experiments consistently showed a thermally-activated transition from $v$-strengthening to -weakening at 80-100°C (Fig. 5a, b), while results from fault healing ('slide-hold-slide' or SHS) tests pointed to an important role for the presence of (pressurized) pore water (Fig. 5a-insets). Specifically, dry samples exhibited classical 'Dieterich-type' healing behaviour (Dieterich, 1978), characterized by a transient peak in shear resistance after each hold period with no effects on steady-state frictional strength (Chen et al. 2015b, c). By contrast, wet experiments showed i) an

increase in apparent steady-state friction upon re-sliding after a hold period, and ii) a pronounced increase of $(a–b)$ after the



SHS stage. Using the hydrothermal ring shear machine Verberne et al. (2015) extended the temperature range to 600°C, which demonstrated a three-regime trend in *(a-b)*-values with increasing temperature reminiscent of that discussed above for phyllosilicate-quartz gouges (Den Hartog and Spiers, 2013; see inset Fig. 5b). Chen et al. (2020a) demonstrated a striking consistency between experimental data spanning the frictional-to-viscous transition in simulated calcite gouge and a

microphysical model that is based on the foundations laid by Bos, Niemeijer, and Spiers (described in detail below).

Regardless of the large changes in *(a-b)* observed in LVF experiments on simulated calcite gouge, the microstructures formed at temperatures up to 550 °C consistently showed localization into at least one, narrow (<100 µm), boundary-parallel shear band. At low temperatures (<150 °C), boundary shears represent a porous, sheet-like volume of calcite nanocrystallites (grain size down to 5 nm) that are locally arranged in dense patches composed of ~100 nm wide spherical grains and fibres (Fig. 5c-

inset) (Verberne et al., 2014a, b, 2019). Towards higher temperatures (400-550 °C) the shear band is composed of linear, cavitated arrays of polygonal grains (~0.3-1 µm in size), suggestive of incomplete grain boundary sliding and (possible post-test) grain growth (Fig. 5d) (Verberne et al., 2017; Chen et al., 2020a). The nano- and microcrystalline boundary shears developed in experiments carried out at ≤550 °C showed a strong crystallographic preferred orientation (Verberne et al., 2013, 2017). Somewhat surprisingly, the post-mortem calcite gouge microstructures resemble those formed in HVF experiments

using simulated gouge material of similar compositions (Smith et al., 2013; De Paola et al., 2015; Rempe et al., 2017; Pozzi et al., 2019). Above 550 °C, a more homogeneous (non-localized), plastically deformed and recrystallized microstructure is observed, consistent with flow controlled by dislocation and possibly diffusional deformation mechanisms (Verberne et al., 2015, 2017; Chen et al., 2020a).

## 5 The Chen-Niemeijer-Spiers microphysical model for shear of gouge-filled faults

Inspired by the modelling work by Bos, Niemeijer, Den Hartog, and Spiers (Figs. 3, 4), and the observations on monomineralic calcite gouge (Fig. 5), Chen and Spiers (2016) developed a more general microphysical model for (localized) shear deformation of gouge-filled faults. This model employs rate-strengthening grain boundary friction plus standard equations for pressure solution creep, and covers both steady state and transient gouge shearing behaviour. It is capable of reproducing results from *v*-stepping as well as SHS tests, using physically-based, independently measured input parameters. In the following, we

describe the main features of this "Chen-Niemeijer-Spiers" (CNS) model.

### 5.1 Model outline

The CNS model assumes an idealized geometry for fault gouge that consists of a densely packed, 2-dimensional array of cylinders or spheres, while allowing for localized deformation in a boundary-parallel shear band located at the margin of a bulk gouge zone (Fig. 6). The shear band and the bulk sample are represented by the same grain packing geometry, but with

different nominal grain diameters internally. Deformation of the gouge layer is controlled by parallel processes active within each zone, specifically, dilatant granular flow plus a creep mechanism such as intergranular pressure solution (IPS). The model



considers frictional energy dissipation for a constant grain size and shear band thickness, ignoring processes such as (de-)localization, grain rolling, or comminution (cataclasis, microcracking). The underlying constitutive equations are derived from kinematic, energy/entropy considerations (Chen and Spiers, 2016)

$$\tau = \sigma_n^{eff} \frac{\tilde{\mu} + \tan(\psi)}{1 - \tilde{\mu}\tan(\psi)} \tag{5a}$$

$$\tilde{\mu} = \tilde{\mu}^* + a_{\tilde{\mu}} \ln\left(\frac{\dot{\gamma}_{gr}}{\dot{\gamma}_{gr}^*}\right) \tag{5b}$$

$$\frac{\dot{\varphi}}{(1-\varphi)} = -\dot{\varepsilon}_{pl} + \dot{\gamma}_{gr}\tan\psi \tag{5c}$$

where $\tau$ is shear stress, $\sigma_n^{eff}$ the effective normal stress, and $\varphi$ shear band porosity. Further, $\dot{\gamma}$ and $\dot{\varepsilon}$ are respectively the shear and normal strain rates, where the subscripts "*pl*" indicates time-dependent ductile (plastic) creep and "*gr*" granular flow.

Equation 5a represents the "friction law" of the CNS model, in which shear resistance is expressed in terms of grain boundary friction $\tilde{\mu}$ and the resistance due to intergranular dilatation $\tan(\psi)$, where $\psi$ is the mean dilatancy angle of the shearing grain pack (Fig. 6). An intrinsically rate strengthening, cohesionless grain boundary slip criterion is adopted (eq. 5b), where $a_{\tilde{\mu}}$ is a strain rate-dependent coefficient and $\tilde{\mu}^*$ is the grain boundary friction coefficient at a reference shear strain rate $\dot{\gamma}_{gr}^*$. Equation 5c captures the evolution of porosity and deformation in the fault-normal direction (i.e., volumetric strains). Granular flow

implies dilatation, or $\dot{\varepsilon}_{gr} = -\dot{\gamma}_{gr}\tan(\psi)$ (Paterson, 1995; Gudehus, 2011). By contrast with the classical RSF equations, the evolving "state" variable in the CNS model, porosity (eq. 5c), is physically measurable and microstructurally quantifiable. To capture transient frictional behaviors, such as occurring upon a perturbation in displacement rate, the deforming gouge plus elastic surrounding (testing apparatus or host rock) is modelled as a spring-slider system assuming zero inertia. Recently, Chen et al. (2019) extended the CNS model to seismic slip rates (~1 m/s), incorporating superplastic flow activated by frictional

heating (De Paola et al., 2015). This refined model is capable of predicting not only low velocity frictional behavior but also (the transition to) rapid dynamic weakening effects frequently seen in high-velocity friction experiments (Di Toro et al., 2011).

**5.2 Comparison with lab data and model predictions**

The CNS model has been successful in reproducing the mechanical behaviours observed in laboratory fault slip experiments (Figs. 7a, b) (Chen and Spiers, 2016; Chen and Niemeijer, 2017; Chen et al., 2017, 2020a, b; Hunfeld et al., 2019, 2020). In

order to reproduce data, the parameters appearing in equations 5a-c are either implicit from the testing configuration used, experimentally-derived values, or else values derived from postmortem microstructural analysis. The model shows favorable consistency with laboratory observations, predicting a dependence of the steady-state friction coefficient on sliding velocity, including a transition from *v*-weakening to -strengthening with increasing *v* (Fig. 7a). Transient strength data upon imposed steps in sliding velocity are also reproduced well, which demonstrates the capability of the CNS model to reproduce the sign

and magnitude of *(a-b)*-values, as well as decaying strength oscillations (Fig. 7b - see the step from 1 to 0.1 µm/s). When



applied over a wide range of sliding velocities, the CNS model output for the steady-state friction coefficient represents a flow-to-friction profile (Fig. 7c), characterized by transitions with increasing $v$ from $v$-strengthening, to -weakening, back to $v$-strengthening, and finally $v$-weakening associated with dynamic, thermal weakening at high velocities (Chen and Niemeijer, 2017; Chen et al., 2019). These $v$-dependence transitions are accompanied by marked changes in mean porosity (bottom **Fig. 7c**). In the intermediate velocity weakening regime, the mean porosity increases with increasing $v$ to relatively high values, however, this decreases to much lower levels when creep becomes the dominant deformation mechanism. Since decreasing shear strain rate in the model is, to some extent, equivalent to an increase in temperature (Bos and Spiers, 2002; Tenthorey and Cox, 2006; Den Hartog and Spiers, 2013), the model can also be used to predict $v$-dependence transitions with increasing temperature.

The above critical sliding velocities or temperatures that mark transitions in fault gouge $v$-dependence can be obtained theoretically. Moreover, the CNS model can be used to derive analytical solutions for the RSF parameters $a$, $b$, and $d_c$, as functions of fault gouge material properties (e.g., solid solubility, activation volume), microstructural parameters (grain size, porosity, shear band thickness), and experimental conditions (temperature, effective normal stress, imposed slip rate) (Chen et al., 2017). This shows that $d_c$ scales with shear band thickness but varies only slightly with slip velocity, whereas the equivalent slip distance in response to large perturbations ($d_0$) increases with the size of the velocity perturbation (Fig. 8a). Using the CNS model as a basis, Van den Ende et al. (2018) showed that the process zone size ($h_{cr}$) for a propagating rupture is $h_{cr} = d_c G / b \sigma_n$, where $G$ is the shear modulus of the elastic medium. This result is consistent with the process zone size or 'nucleation length' $L_b$ derived from RSF analyses (Rubin and Ampuero, 2005). The common foundation of the RSF and CNS models becomes even more apparent when using classical fault stability theory (see eq. 4, Ruina, 1983; Gu et al., 1984). In the CNS model, varying the stiffness $K$ enables simulation of a wide range of transient frictional behaviors frequently seen in laboratory experiments (Baumberger et al., 1994; Leeman et al., 2016), including stable sliding, healing, attenuating or self-sustained (quasi-static) oscillations, and stick-slip (Fig. 8b) (Chen and Niemeijer, 2017). The transition from stable to unstable behavior occurs at the critical stiffness $K = K_c$. The model similarly implies that the transition at low shearing rates, from fully ductile $v$-strengthening behavior to dilatant $v$-weakening (first peak Fig. 7c) marks the point at which $v$-weakening causes acceleration and ultimately a fully dynamic instability (second peak in Fig. 7c).

Interestingly, periodic stick-slip of the type predicted by the CNS model can also be reproduced by numerical discrete element modelling (DEM) of gouges subject to direct-shear deformation that is accommodated by concurrent granular flow and IPS (Van den Ende and Niemeijer, 2018). Even though the detailed mechanics and assumptions used in DEM are dramatically different from those underlying the CNS approach, the concept of shear strain accommodation involving a competition between dilatant granular flow and IPS seems sufficiently robust to reproduce a wide range of frictional sliding modes.





# 6 Earthquake cycle simulations using the CNS model

## 6.1 Empirical and physically-based earthquake cycle simulations

Laboratory observations of fault rock deformation are essentially measurements of a point along a fault that is characterized by a state of stress and thermodynamic conditions. Analytical models such as the CNS model offer a quantitative description

of the mechanical behavior of the fault at that point. Numerical simulations are indispensable for up-scaling these 'point measurements' to the scale of the Earth's crust. In simulations of earthquake rupture nucleation and dynamic propagation, a section of crust or fault is usually discretized such that the continuum is represented by a collection of points, the behavior of each of which is described by a constitutive relation. Over the last few decades, the rate-and-state friction model has been the preferred choice for numerical simulations of fault slip (e.g., Tse and Rice, 1986; Lapusta and Rice, 2003; Thomas et al., 2014;

Luo and Ampuero, 2018). Analyses of these simulations has led to important insights into the spectrum of fault slip behavior (Shibazaki, 2003; Hawthorne and Rubin, 2013), earthquake rupture propagation and arrest (Tinti et al., 2005; Noda and Lapusta, 2013; Lui and Lapusta, 2016), and the relation between the earthquake source and seismological and geodetic observations (Kaneko et al, 2010; Thomas et al., 2017; Barbot, 2019; Ulrich et al., 2019). A major challenge that remains, however, is relating laboratory data of RSF parameters to fault rheology at depth in Earth's crust.

Physics-based models for fault-slip such as the CNS model provide a transparent origin of the constitutive parameters used, so that when employed in numerical simulations, a substantial portion of epistemic uncertainty is eliminated. Conveniently, in the case of the CNS model, its numerical implementation into a seismic cycle simulator (QDYN in the study of Van den Ende et al., 2018) is similar to that of the RSF equations, implying that it is compatible with existing codes for seismic cycle and dynamic rupture simulations. Furthermore, the modular nature of the CNS model enables specific micro-scale deformation

mechanisms to be incorporated, based on microstructural observations of lab-deformed and natural samples. It therefore more closely approaches 'reality', while offering a framework for studying the interaction between time-sensitive and -insensitive deformation mechanisms (i.e., resp. creep and granular flow) that operate in the fault core – as frequently identified from field observations (e.g., Jefferies et al., 2006; Takeshita and El-Fakharani, 2013; Wallis et al., 2015).

## 6.2 Insights into the physics of fault behavior from CNS-based simulations

Because the dynamics of the CNS model are different from rate-and-state-friction (RSF), numerical simulations of fault deformation may lead to new insights into the physics of fault deformation. An example is given by Van den Ende et al. (2018), who reported on a comparison between CNS- and RSF-based seismic cycle simulations. One of their main findings was that even though fault strength evolution near steady-state is practically identical, the behavior far from steady-state is dissimilar, which points to differences in the predicted seismic cycle behavior. Specifically, in the absence of high-velocity dynamic

weakening mechanisms, instead of producing seismic events with large co-seismic slip as expected from RSF-based simulations, CNS-based simulations produce slow-slip events or earthquakes with limited co-seismic displacement (Fig. 9). This can be explained by the transition from $v$-weakening to -strengthening with increasing slip rate, which is implicit in the



CNS model (see Fig. 7c). This transition from *v*-weakening to -strengthening, which has previously been speculated upon by Rubin (2011), emerges naturally from the CNS model, and has the effect of slowing down dynamic ruptures. This, then, constitutes a potential mechanism for the generation of slow earthquakes (see Bürgmann, 2018) for a wide range of rheological parameters and boundary conditions.

The emergence of slow ruptures in numerical simulations is closely related to the nucleation of a frictional instability, which, in the CNS model, occurs near the transition from the ductile creep regime (*v*-strengthening) to the dilatant granular flow regime (*v*-weakening) (Fig. 7c). In numerical simulations that employ classical RSF and an aging law (eqs. 1 and 2a), a rupture becomes dynamic when it exceeds a length scale that is proportional to the nucleation length $L_b$ (Ampuero and Rubin, 2008; Rubin, 2008). When $L_b$ approaches the size of the fault, the rupture is unable to fully accelerate to co-seismic slip rates, causing a slow slip event or slow earthquake (Rubin, 2008). For the CNS model, equivalent expressions for $L_b$ can be obtained that apply to rupture nucleation near the ductile-frictional transition. However, in the case of a transition from *v*-weakening to *v*-strengthening, the transition from slow slip to fast slip is no longer accurately described by traditional estimates of the nucleation length using constant, rate-independent coefficients. Rather, a more detailed fracture mechanics approach (as adopted by Hawthorne & Rubin, 2013, for a modified RSF framework) may shed new light on the parameters controlling earthquake and slow slip nucleation, as well as of the thermodynamic and rheological conditions that control the spectrum of slip modes observed in nature.

Another example that highlights the major benefits of using a physically-based constitutive relation in numerical simulations is in studies of fluid-pressure stimulation of faults. In the case of fluid injection or extraction, the change of hydrological properties with time is crucial for modelling thermo-hydro-mechanical coupling in the system. For the RSF framework, empirical formulae have been proposed aimed at describing the relation between volumetric deformation and $\dot{\theta}$ (Segall and Rice, 1995; Shibazaki, 2005; Sleep, 2005; Samuelson et al., 2009). However, such a relation is not evident from the classical physical interpretation of $\theta$ as an asperity contact life-time. In the CNS model, gouge porosity assumes the role of the state parameter, implying that its evolution can be directly related to changes in fluid pressure, effective normal stress, and/ or hydrological properties within the fault (Van den Ende et al., 2019). To illustrate this, we simulate the evolution of porosity during nucleation, propagation, and arrest of a slow earthquake rupturing a one-dimensional fault with uniform frictional properties (**Fig. 10**), for the regime in which dynamic high-velocity slip is not yet triggered. During the nucleation stage, the fault dilates and weakens simultaneously with accelerated slip. As the rupture reaches its peak slip rates, the gouge attains maximum dilatancy and minimum strength, after which the gouge compacts upon deceleration and rupture arrest. During this cycle of nucleation, propagation, and arrest, the hydrological properties (i.e., hydraulic conductivity) can be computed based on the local porosity. In turn, and informed by laboratory experiments, this enables investigation of the dynamic coupling between fluid flow and fault slip (e.g., Cappa et al., 2019).



## 7 Remaining challenges

To date, the microphysical and earthquake cycle modelling work described above mainly focused on the inter-seismic and nucleation stages of the seismic cycle. For a complete and self-consistent description of fault deformation, co-seismic slip rates must be considered as well. The present model assumptions are reasonable for gouge shear deformation at low slip rates, but break down when frictional heating and associated dynamic fault rupture processes come into play. Specifically, the model requires adaption to include heat production during deformation at ultra-high shear strain rates ($\gg$100 s$^{-1}$), capable of

triggering weakening processes such as thermal pressurization, decomposition, or melting (Rice, 2006; Di Toro et al., 2006, 2011; Platt et al., 2015). As described earlier in this paper, a first step in this direction has been made by Chen et al. (2019), who take into account slip rate-dependent heat production coupled with temperature and grain size sensitivity of creep processes (see Fig. 7c) (cf. De Paola et al., 2015; Pozzi et al., 2019).

Another major challenge yet to be addressed in fault deformation models lies in capturing the dynamics of micro- and

nanostructure formation in sheared fault rock. The CNS model adopts a constant granular structure (Fig. 6), implying that the thickness of the deforming zone, and the grain size within, must be defined a priori. This is problematic for example under conditions close to the transition with increasing strain rate or decreasing temperature from *v*-strengthening to *v*-weakening (ref. Fig. 7c). Constant-*v* experiments on simulated calcite gouge conducted at 550 °C showed that this transition is characterized by progressive shear strain localization from relatively distributed flow involving creep of microcrystalline bulk

gouge grains to granular flow in localized bands, involving much finer grains (<1 μm, see Chen et al., 2020a). Ideally, shear strain localization and grain size reduction are incorporated into a single, fully coupled gouge shear deformation model.

Adequate modelling of granular system dynamics brings with the additional complexity that the deformation properties of individual mineral particles can change with changing particle size. Constraining this is especially important in the case of nanometric gouge (grain size <100 nm), which, along with (partly) amorphized host rocks, are widespread in natural and

experimentally-sheared fault gouges (Power and Tullis, 1989; Yund et al., 1990; for a recent review see Verberne et al., 2019). Individual nanoparticles and nanocrystalline aggregates frequently exhibit dramatically different physical properties compared with their bulk counter-parts (e.g., Meyers et al., 2006; Hochella et al., 2019) —the room temperature ductile nanofibers encountered in calcite gouge being an example of this (Fig. 5c-inset) (Verberne et al., 2014b, 2019). The implication is that extrapolation of data from compaction experiments using micron-sized crystals or larger, used to constrain parameter values

appearing in the CNS model, may lead to large errors when applied to nanogranular or (partly) amorphous fault rock.

Even when coarser grained fault gouges are considered, frictional sliding on the contact between two grains is ultimately governed by nanometer-scale processes. This can either be envisioned as lattice-scale, solid-solid interactions for "dry" contacts or contacts with incomplete coatings of adsorbed species such as water (such as in the CNS model - Chen and Spiers, 2016), or as interactions arising from the unique properties of fully developed, adsorbed water or hydration layers (Leng and

Cummings, 2006; Sakuma et al., 2018). Since the nanometric realm is inaccessible by standard observation techniques, directly probing the processes leading to grain-scale friction remains challenging, in particular for the LVF tests described in **Section**



**3**. Instead, Atomic Force Microscopy (AFM) experiments, also known as Friction Force Microscopy (FFM; see Bennewitz, 2005 for a review), may provide critical observations of the sample response to variations in sliding rate, normal stress, and chemical environment (Diao and Espinosa-Marzal, 2018, 2019). These observations will inform nanophysical models in a

similar way as grain- and aggregate-scale observations have informed the CNS model.

Finally, we note that the capability of the CNS model to fully describe the frictional behavior of strongly heterogeneous gouge compositions, including transients, remains to be investigated. Presently, this has only been demonstrated for monomineralic calcite (Chen and Spiers, 2016; Chen et al., 2017, 2020a). However, in the case that phyllosilicates constitute large portions of the fault gouge the overall constitutive behavior can no longer be represented by taking bulk mean values of rheological

properties (pressure solution kinetics, grain size, etc.). Instead, the interactions between the various phases within the gouge need to be considered more closely, in the assumed microphysical model geometry (Den Hartog and Spiers, 2014 - see **Fig. 4b**), or else using numerical simulations that enable aggregate heterogeneity (i.e., discrete element modelling such as Van den Ende and Niemeijer, 2018; Wang et al., 2019 or finite element methods such as Beall et al., 2019). While microphysical modelling of heterogeneous systems poses some challenges, its potential outcomes likely offer new insights on natural fault

deformation, including on the problem of upscaling to more realistic fault geometries (see e.g., Stenvall et al., 2019).

## 8 Conclusions

We reviewed experimental and microphysical modelling work on the physics of low-velocity fault friction processes, carried out at Utrecht University (UU) since the early 2000's. Data from shear deformation experiments on simulated fault rocks composed of halite-phyllosilicate and phyllosilicate-quartz mixtures, and of monomineralic calcite, consistently show that fault

gouge strength and stability is controlled by a competition between rate-sensitive creep and rate-insensitive granular flow processes. Under conditions where ductile deformation occurs in Earth's crust, fault shear deformation is non-dilatant and controlled purely by creep, which is intrinsically stable. However, towards shallower depths, frictional(-viscous) deformation occurs, which is controlled by creep of individual mineral grains operating alongside dilatant granular flow. The seismogenic zone represents a depth interval in the crust where these processes operate at comparable rates, $|\dot{\varepsilon}_{dil}| \approx |\dot{\varepsilon}_{cp}|$, which leads to

velocity weakening hence seismogenic fault-slip behavior. This conceptual model framework is quantitatively described by the Chen-Niemeijer-Spiers (CNS) model for shear of gouge-filled faults, which constitutes a physically-based microphysical model that is capable of reproducing a wide range of (transient) frictional behaviors. Implemented into numerical codes for fault rupture, the CNS model offers new, microstructurally and physically founded input for earthquake cycle simulators, and therewith new scope for the interpretation of earthquake source processes.



## Availability of data and materials

All data are available from the papers cited, or else upon request from the corresponding author. The QDYN seismic cycle simulator (including the implementation of the CNS model) is open-source available from https://github.com/ydluo/qdyn.

## Competing interests

The authors declare no conflicts of interest.

## Funding

BAV acknowledges support from JSPS KAKENHI grant #19K14823, and MPAvdE from the French government through the UCA[JEDI] Investments in the Future project managed by the National Research Agency (ANR) (reference # ANR-15-IDEX-01). JC acknowledges support by the European Research Council under grant SEISMIC (335915), awarded to ARN.

## Author contributions

BAV led the effort, drafted the initial manuscript, and wrote sections 1 and 3. BAV and MPAvdE co-wrote sections 2, 4, 7, and 8, with contributions by JC to section 4. JC wrote section 5, and MPAvdE section 6. All authors contributed to writing of the final manuscript

## Acknowledgements

The research work described in this paper would have been impossible without the support from Magda Mathot-Martens, Eimert de Graaff, Gert Kastelein, Thony van der Gon-Netscher, Peter van Krieken, Otto Stiekema, Leonard Bik, Floris van Oort, and Gerard Kuijpers. Colin Peach is thanked for his major scientific contributions throughout the years, as well as for developing, improving, and maintaining experimental equipment. Miki Takahashi is thanked for providing the photo in Figure 2a.

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





**Figure 2. Apparatuses and testing assemblies used for low-velocity, ring- and direct-shear friction experiments at UU. a) and d) the room temperature ring-shear set-up. b and e) the hydrothermal ring-shear set-up. c and f) the direct-shear set-up, using a triaxial pressure cell (in this case the "shuttle vessel"). Photo (a) by courtesy of Miki Takahashi.**
870 **Photo (d) taken from Van den Ende and Niemeijer (2019).**





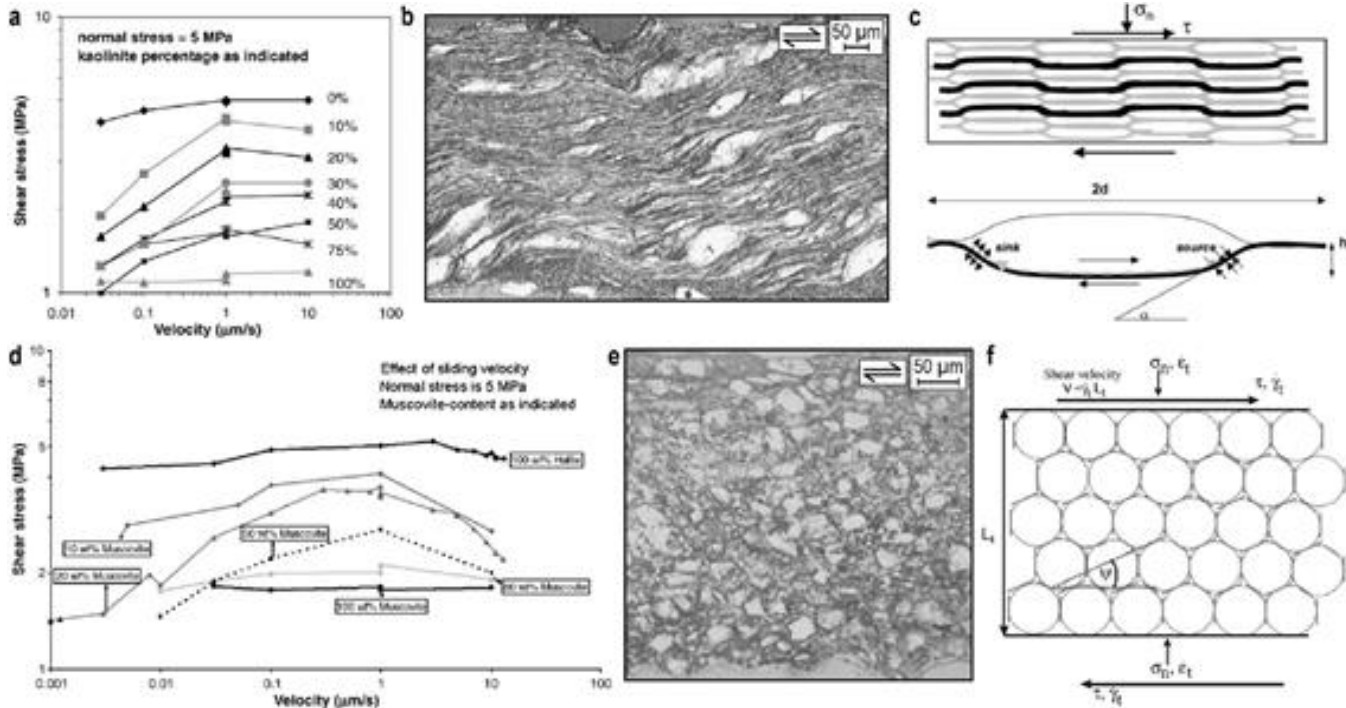

**Figure 3. Key results from room temperature ring-shear experiments on halite-phyllosilicate mixtures. a) Velocity (*v*)-strengthening of halite-kaolinite gouges, and b) micrograph of a sample deformed in the v-strengthening, frictional-viscous regime (both from Bos et al., 2000b). c) Model framework of Bos and Spiers (2002a). d) Velocity-dependence effects in halite-muscovite gouges (from Niemeijer and Spiers, 2006). e) Micrograph of a sample deformed in the *v*-weakening regime (from Niemeijer and Spiers, 2005). d) Model framework of Niemeijer and Spiers (2007). All images taken with publishers' permission.**





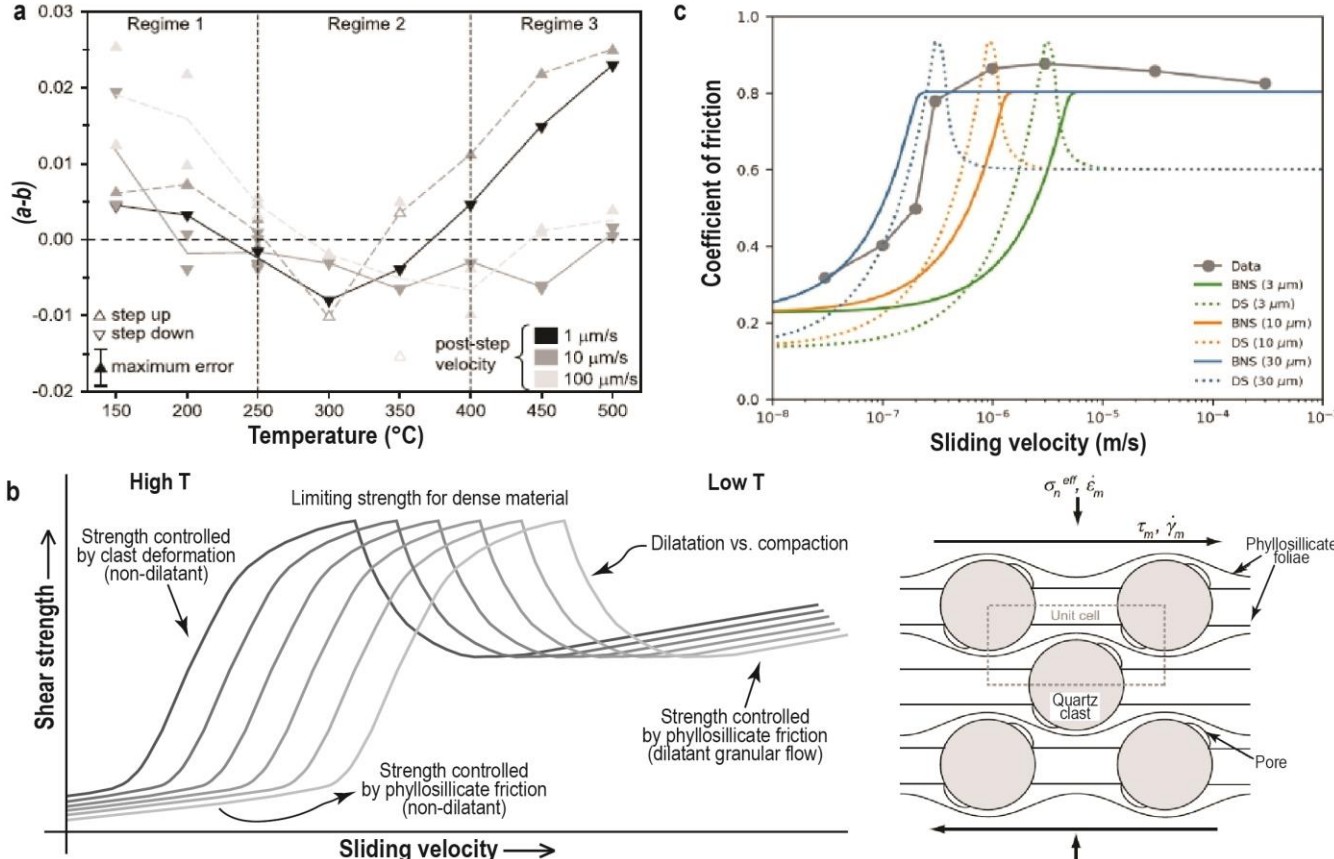

**Figure 4. Key results from hydrothermal ring-shear experiments on quartz-phyllosilicate mixtures. a) Three-regimes of *v*-dependence seen in experiments on illite-quartz mixtures (from Den Hartog and Spiers, 2014) . b) Interpretation of quartz-phyllosilicate strength evolution with sliding velocity/ temperature (after Den Hartog and Spiers, 2013). The right-hand side shows the model microstructure envisioned by Den Hartog and Spiers (2014) (DS). c) Plot of the coefficient of friction at steady-state against sliding velocity with data from constant-*v* experiments by Niemeijer (2018). Curves show results of model calculations using the model for frictional-viscous deformation by Bos and Spiers, 2002a and Niemeijer and Spiers, 2005, and using the DS model. All images taken with publishers' permission.**



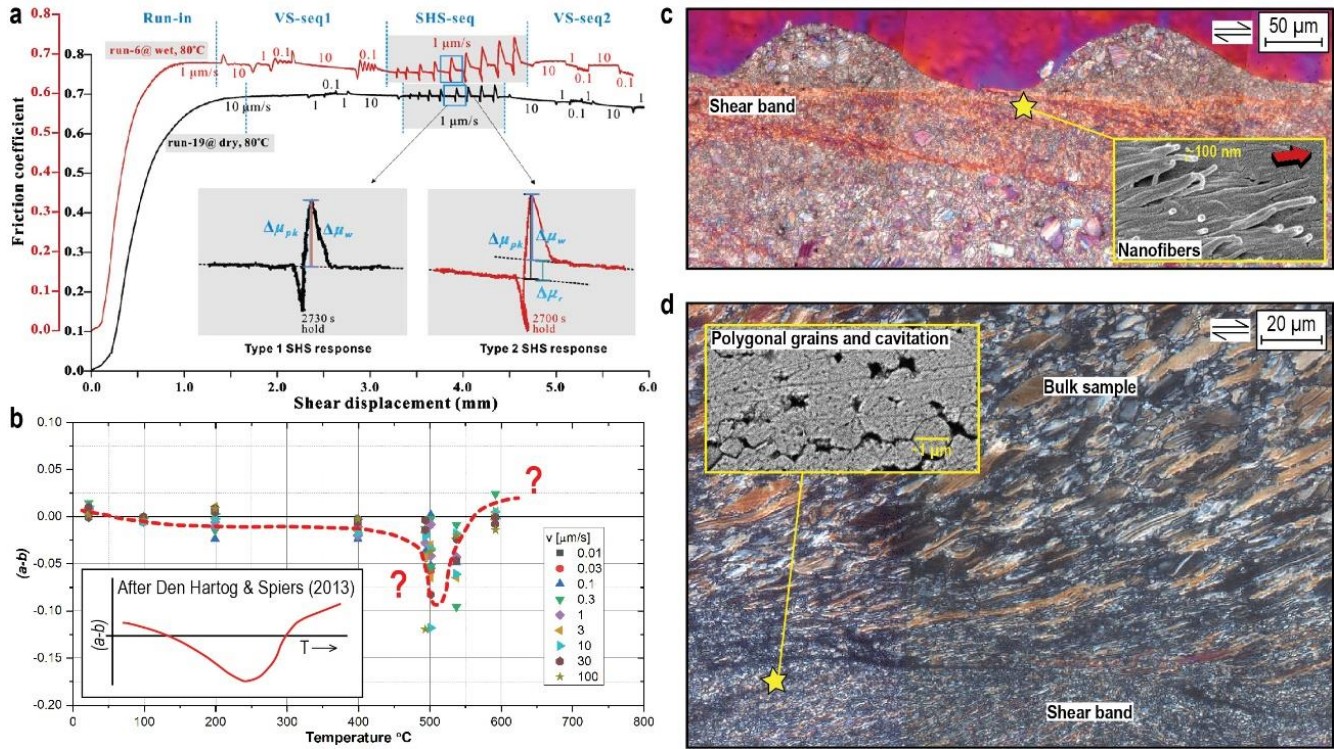

**Figure 5. Key results from experiments on simulated calcite fault gouge. a) Velocity-stepping (VS) and slide-hold-slide (SHS) experiments (from Chen et al., 2015b). b) Data from v-stepping experiments performed up to 600 °C (Verberne et al., 2015). c and d) Cross-polarized, transmitted light micrographs of samples sheared in room temperature *v*-stepping experiments (c) and at 500°C (d) (from Verberne et al., 2013, 2015). Inset in (c) shows nanofibers observed after an experiment. Inset in (d) highlights cavitated arrays of polygonal grains observed in a shear bands developed at high temperatures. All images taken with publishers' permission.**





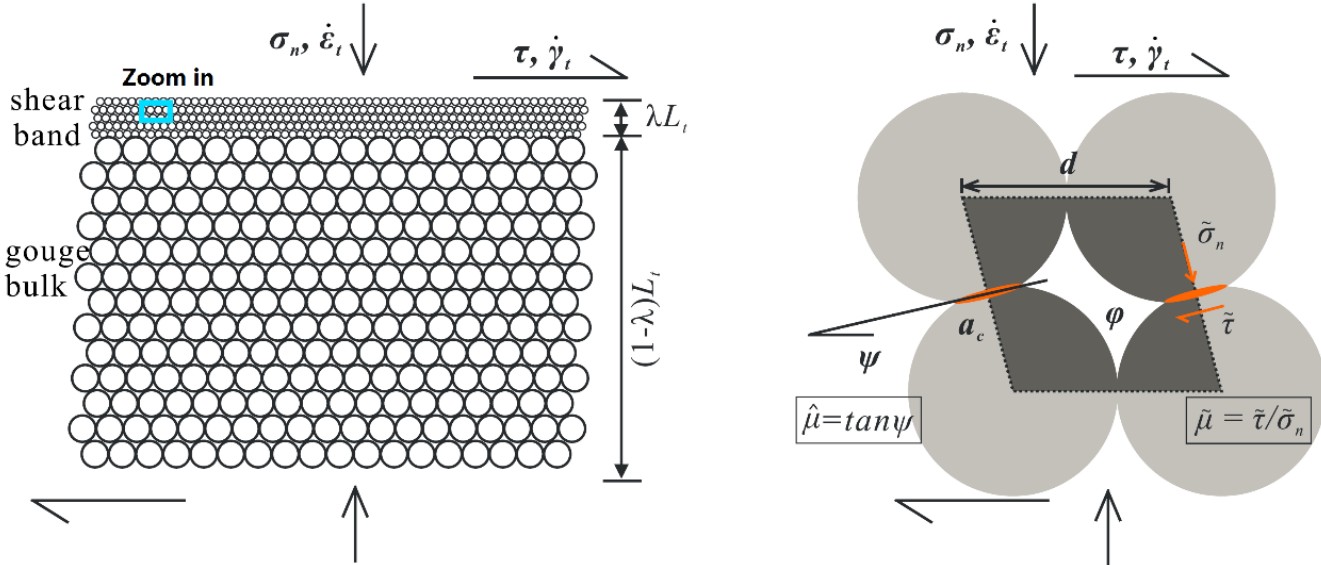

**Figure 6. Gouge layer geometry assumed in the Chen-Niemeijer-Spiers (CNS) model. After Chen and Spiers (2016).**

895




**Figure 7. Examples of CNS model output. Reproduction of a) steady-state frictional strength and *(a-b)* values, and b) *v*-stepping tests, of data from experiments on calcite gouge performed at $T = 80°C$ and $\sigma_n^{eff} = 50$ MPa (see Fig. 5a, Chen et al., 2015b). c) Model output for over a wide range of sliding rates, highlighting a flow-to-friction transition with increasing *v* (see Chen et al., 2020a).**





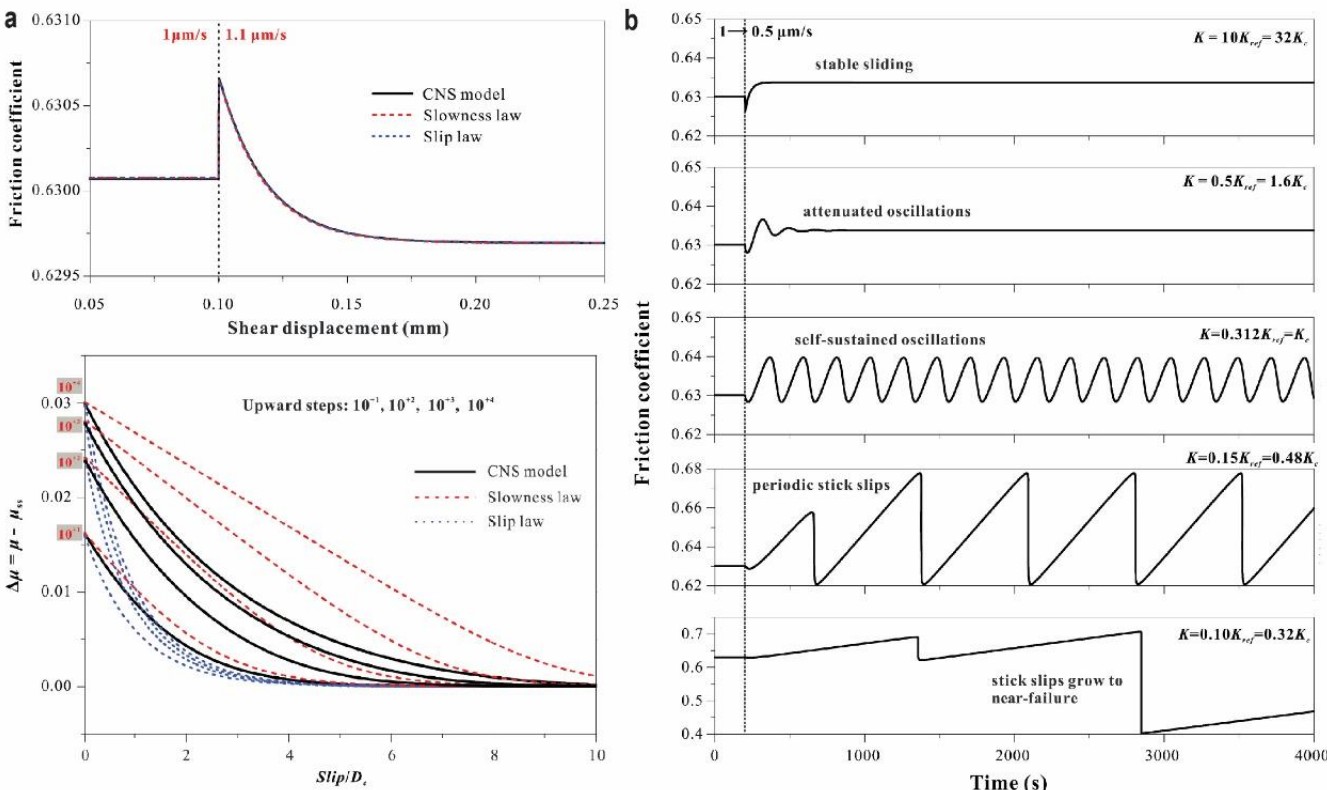

**Figure 8. CNS modelling results. a) Comparison with RSF model for small (1→1.1 µm/s) and large (up to 4 orders of magnitude) perturbations in sliding velocity (resp. the upper and lower diagrams). The RSF parameter values used are the equivalent values calculated from the CNS model (see Chen et al., 2017). b) CNS modelling of a velocity step 1→0.5 µm/s, for different stiffnesses _K_ (see Chen and Niemeijer, 2017).**

905





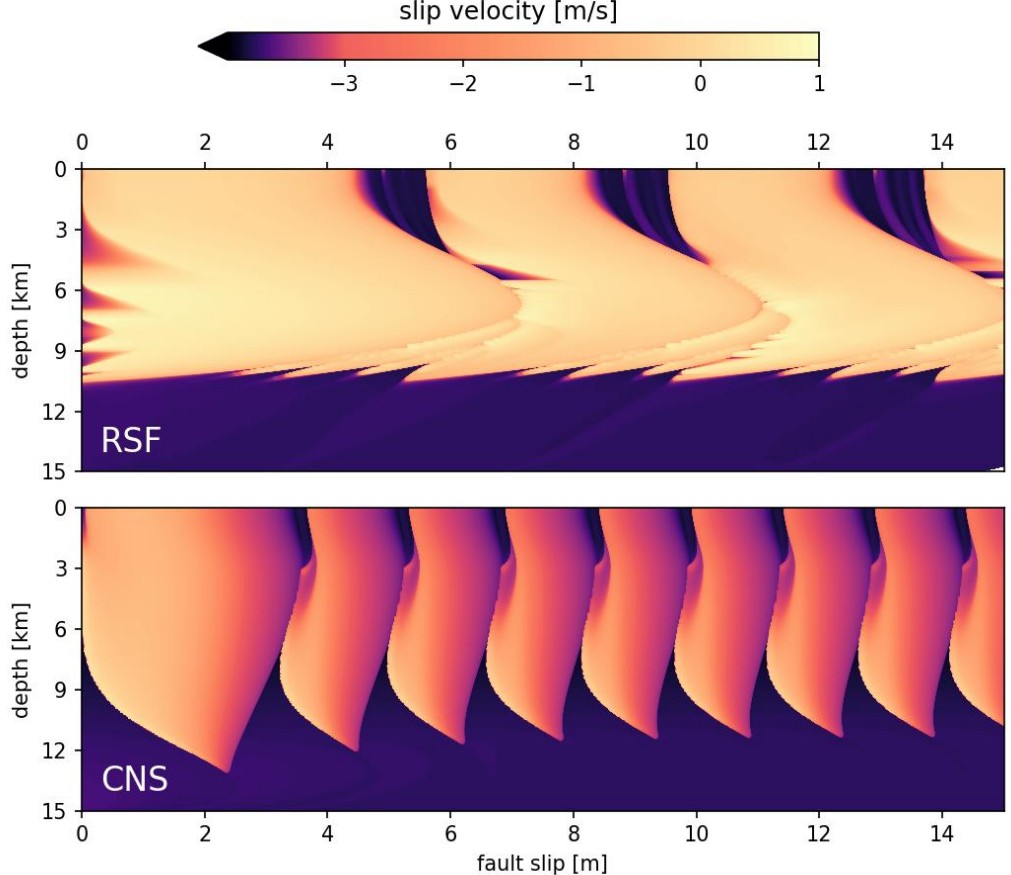

**Figure 9. Numerical simulation of the spatio-temporal evolution of slip rates on a simulated strike-slip fault, based on rate-and-state friction (RSF; top), and on the Chen-Niemeijer-Spiers model (CNS; bottom). Brighter colours indicate higher slip rates, dark purple colors indicate slow creep. While the RSF-based simulation exhibits large and fast earthquakes, the CNS-based simulation exhibits mostly small slow slip events. After Van den Ende et al. (2018).**



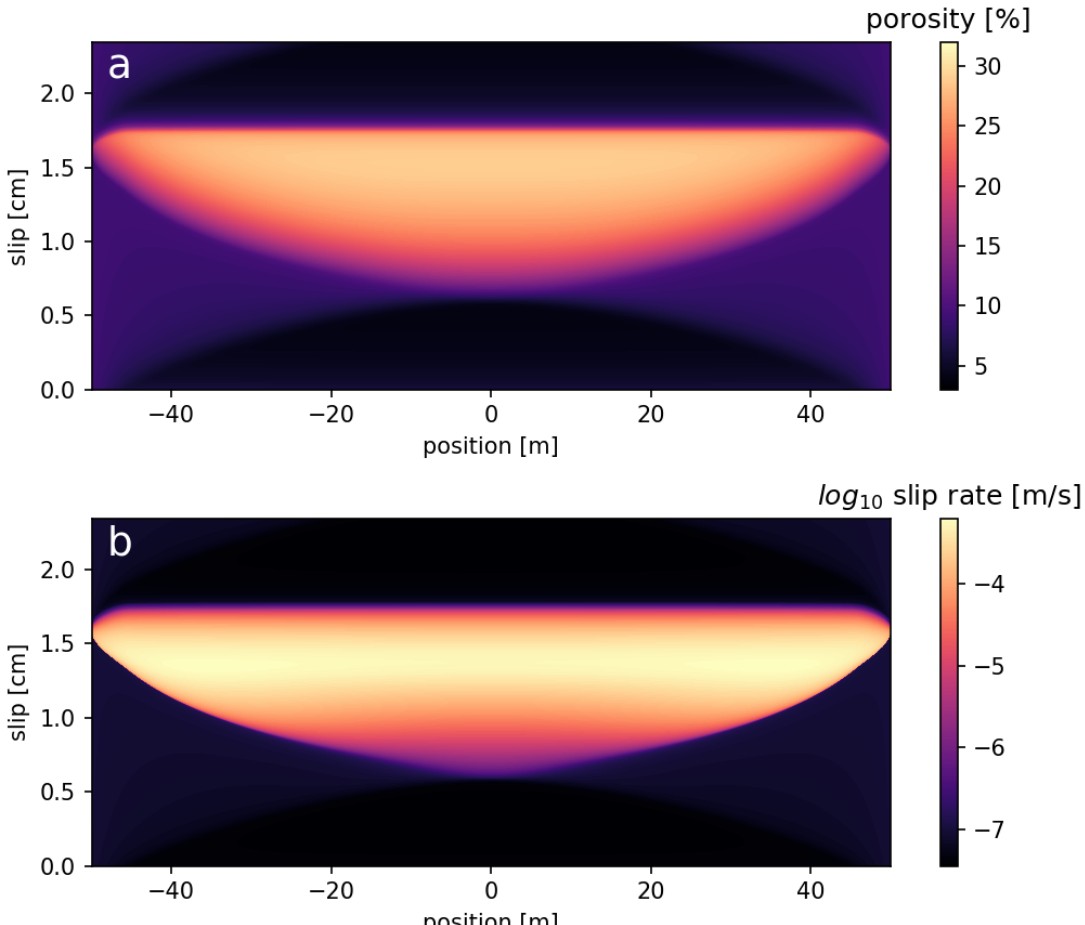

**Figure 10. Spatiotemporal evolution of fault gouge porosity (a) and slip rate (b) during nucleation, propagation, and arrest of a rupture on a fault with uniform frictional properties. The rupture nucleates in the centre and propagates outwards.**