# Peer review of "The physics of fault friction: Insights from experiments on simulated gouges at low shearing velocities"

_Solid Earth, 2020_

## Referee Comment (RC1) · Anonymous Referee #1 · 13 Jul 2020

Dear Editor, I have reviewed the paper entitled "The physics of fault friction: insight from experiments on simulated gouges at low shearing velocities" by Verberne et al., as a potential article to be published in Solid Earth. I start by saying that this is a review article and does not contain any new results. The authors summarize a vast amount of work that that has been carried out in the last 20 years or more at the Utrecht University rock mechanics laboratory. The main focus of this review is to give the reader a general overview of the development, and the current state-of-the-art, including strengths and limitations, of the CNS (Chen-Niemeijer-Spiers) microphysical model for fault friction and the earthquake cycle that is based on experimental evidences derived from experiments performed at the Utrecht laboratory. The review is well organized, in fact

the authors start by clearly describe the experimental facilities used to perform the experiments, then they summarize some case studies that have put the basis for the formulation of CNS model. Afterwards, they describe the theoretical foundation of the model validating it with a comparison with the experimental data. Finally, they discuss the CNS model as applied to earthquakes simulation with a comparison with Rate- and State- Friction (RSF) constitutive equations that is the most used framework so far. Throughout the review, the authors make the appropriate references to direct the reader to the relevant papers that have been published regarding the various aspect of the development of the CNS model. The papers are all in very good journals and highly cited so that the scientific basis for this model is not under discussion. In general the paper is very well written and organized and the figures are appropriate and it represent a necessary step to summarize the work done in developing this model. For these reasons I recommend publication after the authors address some minor concerns as listed below.

Comments to the authors: General: A minor aspect, but it should absolutely improve, regards the figures. From figure 3 throughout figure 10, they look like screen shots taken from the cited articles. This is true to the point that in figure 3 is impossible to read the text in the different figure panels. I strongly recommend the authors to produce high quality figures. I think that one aspect that should be improved in this review, since in the single papers cited is poorly addressed, is the relation between the CNS model and its physical basis with the interpretation of the mechanical work related to dilation as it was developed by Marone et al., 1990 JGR and Beeler et al., 1996 JGR. In particular they interpret the velocity dependence of friction (v-strengthening or weakening) based on an energy balance of the work done against the normal stress (i.e. dilation rate) and relate it with the degree of shear localization. This basis are very similar to the CNS model. However, something that is not very clear to me is that in the observations of Marone and Beeler velocity perturbations that lead to fault dilation are associated with velocity strengthening frictional behavior, and shear localization is associated with a velocity weakening behavior. While in the CNS model increasing in porosity leads to

[Figure]

velocity weakening and "localization" by ductile mechanisms to velocity strengthening. Those experiments were conducted on crystalline material such as quartz or granite and the CNS model was developed for calcite that notably undergoes IPS. Can the authors implement some comments about these models? Specific: L39: "Seismic fault motion of this type" it is not very clear to me. The authors refer to many fault slip styles in the previous sentence, so I would rephrase maybe with "the full spectrum of slip behaviors". L84-91: About the frictional-viscous mechanism. I would avoid to generalize too much such mechanism as active at crustal scale everywhere as it reads now in the text. It is true that from the outcrop observation of ancient subduction zones and some phyllosilicate-rich faults (e.g. Fagereng et al., 2014; Collettini et al., 2011 these references may be added to the text) this behavior can be inferred as at play. However, this is not true for all the fault zones and I think that it is not appropriate to generalize it. Alternatively, it should be specified that the seismogenic zone refer only to subduction zones here. L110: since this is a review article I would give credit to the people that put the basis for the friction contact theory such as the work of Bowden and Tabor as well as Rabinowicz and not only Dieterich and Kilgore 1994. L330: in regard to the scaling of the critical slip distance with fault thickness I think that the citation to Marone and Kilgore, 1993 Nature is needed.
* * *

---

## Referee Comment (RC2) · Anonymous Referee #2 · 12 Aug 2020

General comments: This paper summarizes experimental, microstructural, microphysical, and numerical modeling studies of the frictional behavior of simulated gouge conducted at Utrecht University (UU) in the past two or so decades. Although the paper does not have any new results, except some of the modeling results only presented in the recent international conference, it includes the basic information on the experimental setup and results, microphysical model, and numerical modeling of earthquake cycles using their friction model (CNS model). The paper is based on many previous papers presented by UU group and hence it is very well written. Therefore, I recommend publication only after the following minor comments shown below.

[Figure]

Specific comments:

Quality of figures: As mentioned in the reviewer 1, overall quality of figures is too low. Some of the words can't be read. Hence it should be improved.

The difference in the frictional properties. Line 258-inset of Fig. 5b: I assume that the authors try to mention the similarity of temperature dependences between calcite (Fig. 5) and qtz-phyllosilicate mixture (Fig. 4a) gouges. However, their variations as a function of temperature are different. In particular, calcite gouges show a wide region of negative v dependence with sharp peaks at 500 C. Can you elaborate more on the difference? Because as shown in later, the CNS model offers fault friction law based on microphysics supported by those experiments. If the authors can illuminate tho difference in terms of rock and mineral physics aspects, this will give a more generalized view on the microphysics of fault friction. I assume that is a point the CNS model aims.

Robustness of the CNS model As repeatedly mentioned in the paper, CNS model is based on microphysics supported by experimental results. In that sense, the CNS model provides a transparent origin of the constitutive parameters. However, as shown in Fig. 9 and Chapter 7 (Remaining challenges), the CNS model has a significant shortcoming on that it can only reproduce slow slip or earthquakes with limited coseismic displacement. Hence, I guess that the authors should avoid bold statements on the robustness of the model (e.g., lines 25, 361, 468, and so on).

Lines 169-170: "The maximum rotation or shear displacement that can be achieved is limited by the water cooling and pore fluid systems, ". What does that mean? It is better to elaborate more on the experimental detail for readers outside of the field.

Line 240 and Fig. 4: Need a more detailed explanation of the Fig. 4 (in particular 4c). For example, explanations on the differences in color and meaning of peaks in dash-lines are needed.

Line 250: "an important role for the presence of (pressurized) pore water (Fig. 5ainsets) " But how can we understand the importance of ( (pressurized) pore water) from Fig. 5a?

---

## Author Comment (AC1) · 9 Sep 2020

SEE THE SUPPLEMENT FOR OUR REPLIES IN RED

Anonymous Referee #1

Dear Editor, I have reviewed the paper entitled "The physics of fault friction: insight from experiments on simulated gouges at low shearing velocities" by Verberne et al., as a potential article to be published in Solid Earth. I start by saying that this is a review article and does not contain any new results. The authors summarize a vast amount of work that that has been carried out in the last 20 years or more at the Utrecht University

rock mechanics laboratory. The main focus of this review is to give the reader a general overview of the development, and the current state-of-the-art, including strengths and limitations, of the CNS (Chen-Niemeijer-Spiers) microphysical model for fault friction and the earthquake cycle that is based on experimental evidences derived from experiments performed at the Utrecht laboratory. The review is well organized, in fact the authors start by clearly describe the experimental facilities used to perform the experiments, then they summarize some case studies that have put the basis for the formulation of CNS model. Afterwards, they describe the theoretical foundation of the model validating it with a comparison with the experimental data. Finally, they discuss the CNS model as applied to earthquakes simulation with a comparison with Rate and State- Friction (RSF) constitutive equations that is the most used framework so far. Throughout the review, the authors make the appropriate references to direct the reader to the relevant papers that have been published regarding the various aspect of the development of the CNS model. The papers are all in very good journals and highly cited so that the scientific basis for this model is not under discussion. In general the paper is very well written and organized and the figures are appropriate and it represent a necessary step to summarize the work done in developing this model. For these reasons I recommend publication after the authors address some minor concerns as listed below. We thank the reviewer for his/her constructive review, which has helped to improve the quality of the ms.

Comments to the authors: General: A minor aspect, but it should absolutely improve, regards the figures. From figure 3 throughout figure 10, they look like screen shots taken from the cited articles. This is true to the point that in figure 3 is impossible to read the text in the different figure panels. I strongly recommend the authors to produce high quality figures. We fully agree that the figure quality in the .pdf version of the submitted ms was inadequate. We suspect that this may have occurred upon rendering a merged version of the ms. To ensure the best quality figures in the final manuscript, each figure as been re-exported from their original format (.ai or .cdr) to .jpg format, at 300 DPI resolution. In addition, we have increased the overall quality of

each figure by adjusting text within, adopting a uniform style (font type, size, etc.), and by improving the figure captions.

I think that one aspect that should be improved in this review, since in the single papers cited is poorly addressed, is the relation between the CNS model and its physical basis with the interpretation of the mechanical work related to dilation as it was developed by Marone et al., 1990 JGR and Beeler et al., 1996 JGR. In particular they interpret the velocity dependence of friction (v-strengthening or weakening) based on an energy balance of the work done against the normal stress (i.e. dilation rate) and relate it with the degree of shear localization. This basis are very similar to the CNS model. However, something that is not very clear to me is that in the observations of Marone and Beeler velocity perturbations that lead to fault dilation are associated with velocity strengthening frictional behavior, and shear localization is associated with velocity weakening behavior. While in the CNS model increasing in porosity leads to velocity weakening and "localization" by ductile mechanisms to velocity strengthening. Those experiments were conducted on crystalline material such as quartz or granite and the CNS model was developed for calcite that notably undergoes IPS. Can the authors implement some comments about these models? This is an interesting point raised by the reviewer, which we are happy to discuss. The interpretation of the mechanisms controlling fault gouge shear deformation offered by Marone et al. (1990) and Beeler et al. (1996) is rooted in observations of fault strength, velocity dependence, and dilatation, from experiments on simulated quartz(-rich) gouges. The correlations between the data reported are intriguing indeed, and the interpretations offered stimulating -as recognized from the important work that was directly or indirectly inspired by it (see e.g., Segall and Rice, 1995; Scruggs and Tullis, 1998; Sleep et al., 2000; Samuelson et al., 2009). However, the Marone/ Beeler model is based on an intuitive assumption or hypothesis, which, while inspiring and broadly consistent with trends seen in their experimental data, has no explicit physical or thermodynamic origin. Quoting Beeler et al. (1996),"Marone etal. [1990] hypothesized that the friction velocity dependence of simulated gouge is the sum of the friction velocity dependence of bare

surfaces and the velocity dependence of dilation rate". The CNS model re-evaluates the role of contact friction and dilation, using an established energy balance approach. In effect, the CNS model assumes that gouge shear resistance is caused by energy consumed in driving i) grain boundary friction (equivalent to 'bare surface friction' of Marone et al.) and ii) dilatation. However, net (i.e. measured) dilatation, constant in the case of steady-state sliding, is the result of competition between porosity increase caused by rate-insensitive granular flow (characterised by a dilatancy angle) and intergranular compaction by time-dependent creep processes (such as IPS mentioned by the reviewer - though the same principles apply for any creep mechanism). In the Marone/ Beeler model, intergranular creep processes are ignored, which means that dilatation is artificially assumed to be shear rate dependent and always leads to an increase in porosity. If this increase occurs faster than the rate at which pore fluid can flow in, the pore fluid pressure will decrease, leading to an increase in effective normal stress and so-called 'dilatancy strengthening' (Segall and Rice, 1995; Samuelson et al., 2009). While the latter is relevant especially in the case of transients, micro- and nanostructural observations, as well as compaction phenomena occurring during slide-hold-slide experiments, for example, clearly demonstrate an important role of creep mechanisms in friction experiments, suggesting that their incorporation in models for shear of gouge-filled faults represents a key step towards capturing physical reality. Regarding the inferred relation between shear strain localization and velocity weakening behaviour, we further note that there is little microstructural evidence of any systematic nature to support this. When examined with a microscope, sheared gouges usually show a localized shear band structure regardless whether the mechanical data implies velocity strengthening or -weakening behaviour (see Verberne et al., 2013, 2015, 2019; Niemeijer, 2018). However, in view of the major differences between our experiments and those of Marone/Beeler and co-workers (i.e., materials investigated, P-T-v conditions employed), it is perhaps not realistic to expect full agreement of the results. More work is needed on fault gouge microstructure development and its relation with sliding velocity, normal stress, and displacement. Until this is clarified, care should

be taken when interpreting a relation between fault mechanical properties and gouge microstructure. To address the reviewers' comment, in the revised ms we now include a brief statement on the important findings regarding the role of dilatation reported by Marone et al. (1990), and on the subsequent modelling work that was derived from it (lines 138-142). While we acknowledge the seminal impact of the work by Marone et al (1990) and Beeler et al al (1996), we feel that a detailed comparison with the CNS model would be too lengthy to include in the present ms

Specific: L39: "Seismic fault motion of this type" it is not very clear to me. The authors refer to many fault slip styles in the previous sentence, so I would rephrase maybe with "the full spectrum of slip behaviors". OK. We were referring to 'slow slip and earthquakes', mentioned in the preceding sentence. We have now clarified this accordingly by rephrasing (line 40).

L84-91: About the frictional-viscous mechanism. I would avoid to generalize too much such mechanism as active at crustal scale everywhere as it reads now in the text. It is true that from the outcrop observation of ancient subduction zones and some phyllosilicate-rich faults (e.g. Fagereng et al., 2014; Collettini et al., 2011 these references may be added to the text) this behavior can be inferred as at play. However, this is not true for all the fault zones and I think that it is not appropriate to generalize it. Alternatively, it should be specified that the seismogenic zone refer only to subduction zones here. We acknowledge that the wording in the original ms was somewhat over-confident here. Part of the 'problem' is the use of the term 'frictional-viscous' which, through time, has received the connotation referred to by the reviewer, as being associated with subduction megathrusts or phyllosilicate-rich faults. However, our intention is much broader than this. In particular, recent observations demonstrate that concurrent brittle/frictional-plastic deformation is widespread in nanogranular fault rocks. To address the reviewers' point, we now write (lines 85-93)

"Within the seismogenic zone and shallower, field and laboratory observations on a wide range of fault rock types point to the concurrent operation of brittle/frictional

(cataclastic) processes that depend linearly on effective normal stress, and rate-sensitive, plastic deformation processes (e.g., pressure solution, dislocation- or diffusion-mediated creep) (Wintsch et al., 1995; Holdsworth et al., 2001; Imber et al., 2008; Collettini et al., 2011; Siman-Tov et al., 2013; Fagereng et al., 2014; Delle-Piane et al., 2018; Verberne et al., 2019). The relation between this 'frictional-plastic' deformation of fault rock and seismogenesis, including of the competing effects between time-sensitive /-insensitive deformation processes on failure, creep, compaction, and healing….."

We thank the reviewer for the references that he/she mentioned, which we have now included. To strengthen our cause, we also mention here the work by Siman-Tov et al., 2013), Delle-Piane et al. (2018), and the review by Verberne et al. (2019), whom demonstrated the importance of frictional-plastic processes in nanogranular fault rock.

L110: since this is a review article I would give credit to the people that put the basis for the friction contact theory such as the work of Bowden and Tabor as well as Rabinowicz and not only Dieterich and Kilgore 1994. The reviewer is right. We followed his/her suggestion by adding references to Bowden and Tabor (1950, 1964) and Rabinowicz (1956, 1958) (line 111-112).

L330: in regard to the scaling of the critical slip distance with fault thickness I think that the citation to Marone and Kilgore, 1993 Nature is needed. This is an important result and we thank the reviewer for pointing this out. The relation between dc and shear band thickness, which the CNS model independently arrives at, is effectively the same as what Marone and Kilgore (1993) arrived at on the basis of RSF analysis. We now mention this in line 334.

References NOT cited in the revised ms

Scruggs, V. J., & Tullis, T. E. (1998). Correlation between velocity dependence of friction and strain localization in large displacement experiments on feldspar, muscovite and biotite gouge. Tectonophysics 295, 15-40. Sleep, N. H., Richardson, E., and

Marone, C. (2000). Physics of friction and strain rate localizatio

Please also note the supplement to this comment:
https://se.copernicus.org/preprints/se-2020-85/se-2020-85-AC1-supplement.pdf
* * *

---

## Author Comment (AC2) · 9 Sep 2020

SEE SUPPLEMENT FOR OUR REPLIES IN RED

Anonymous Referee #2

General comments: This paper summarizes experimental, microstructural, microphysical, and numerical modeling studies of the frictional behavior of simulated gouge conducted at Utrecht University (UU) in the past two or so decades. Although the paper does not have any new results, except some of the modeling results only presented in the recent international conference, it includes the basic information on the experi-

mental setup and results, microphysical model, and numerical modeling of earthquake cycles using their friction model (CNS model). The paper is based on many previous papers presented by UU group and hence it is very well written. Therefore, I recommend publication only after the following minor comments shown below. We thank the reviewer for his thoughtful comments, which have helped to improve the quality of the ms.

Quality of figures: As mentioned in the reviewer 1, overall quality of figures is too low. Some of the words can't be read. Hence it should be improved. Yes, point taken. We now apply a consistent style between the figures, and ensured a high resolution (300 DPI) upon exporting the figures to JPEG format. We are confident that this will constitute the high figure quality needed for publication. See also our response to the first comment by Reviewer 1.

The difference in the frictional properties. Line 258-inset of Fig. 5b: I assume that the authors try to mention the similarity of temperature dependences between calcite (Fig. 5) and qtz-phyllosilicate mixture (Fig. 4a) gouges. However, their variations as a function of temperature are different. In particular, calcite gouges show a wide region of negative v dependence with sharp peaks at 500 C. Can you elaborate more on the difference? Because as shown in later, the CNS model offers fault friction law based on microphysics supported by those experiments. If the authors can illuminate tho difference in terms of rock and mineral physics aspects, this will give a more generalized view on the microphysics of fault friction. I assume that is a point the CNS model aims. This is a good point raised by the reviewer. Taking the CNS model in mind, the shape of the temperature sensitivity of v-dependence (i.e., plotted in Fig. 4a and Fig. 5b) is expected to be dominantly controlled by the rate of intergranular creep. In the case of calcite, intergranular creep occurs by water-assisted diffusive mass transfer at low temperatures (T<150°C) (Verberne et al., 2014a,b; Chen et al., 2015a,b; Chen & Spiers, 2016), and by dislocation- / diffusion-mediated plasticity at higher temperatures (Verberne et al., 2015; Chen et al., in review). In the case of qtz-pyllosilicate gouges,

the intergranular creep mechanisms are more difficult to constrain, and the modelling is more complex (Den Hartog and Spiers, 2014; Niemeijer, 2018). As pointed out in section 7, one of the key remaining challenges is to quantitatively underpin the relevant creep processes in polymineralic gouges, and their incorporation into the CNS model. This remains subject of future study. Instead of comparing Fig. 4a with Fig. 5b, as mentioned by the reviewer, in fact we wanted to highlight the similarity in the shape of the (a-b) vs T curve with the derivative of the curves in Fig. 4b (sketched in the inset to Fig. 5b), that is, the inherent prediction from the Den Hartog and Spiers' (2013) model. We realize that this may have been confusing. To address this we now more specifically mention the comparison between Fig. 4b and Fig5b-inset (lines 263-266).

Robustness of the CNS mode. As repeatedly mentioned in the paper, CNS model is based on microphysics supported by experimental results. In that sense, the CNS model provides a transparent origin of the constitutive parameters. However, as shown in Fig. 9 and Chapter 7 (Remaining challenges), the CNS model has a significant shortcoming on that it can only reproduce slow slip or earthquakes with limited coseismic displacement. Hence, I guess that the authors should avoid bold statements on the robustness of the model (e.g., lines 25, 361, 468, and so on). OK, point taken. When the reviewer mentions that 'the CNS model can only reproduce slow slip or earthquakes with limited coseismic displacement', we assume that he/she is referring to the incorporation of dynamic weakening processes at co-seismic slip rates, or lack thereof. This is indeed the case in its present form (see Section 5), however, the first steps to a unified model are already under way (poster by Chen et al. at GeoProc international conference, 2019). To address the reviewers' comment, we have rephrased statements on the robustness of the CNS model. We included notes that refer to the challenges ahead (see lines 25-26, 371-374, 468-469).

Lines 169-170: "The maximum rotation or shear displacement that can be achieved is limited by the water cooling and pore fluid systems, ". What does that mean? It is better to elaborate more on the experimental detail for readers outside of the field. The
water cooling and pore fluid systems include connections (i.e., hoses and tubing) to external reservoirs. These connections must be able to accommodate rotation of the vessel. Smart designs have helped to accommodate rotation such that very large sample displacements can be achieved (>100 mm). Following the reviewers' suggestion, we rephrased line 176-177 in accordance with the above.

Line 240 and Fig. 4: Need a more detailed explanation of the Fig. 4 (in particular 4c). For example, explanations on the differences in color and meaning of peaks in dash-lines are needed. OK. We have improved the overall quality of Figure 4 (see reply to first comment), including readability of Fig. 4c. We rewrote the caption to ensure that all abbreviations, symbols, colours, etc used are explained (applies to all figures), and we included a description on the meaning of the peaks in Fig. 4c.

Line 250: "an important role for the presence of (pressurized) pore water (Fig. 5a-insets) But how can we understand the importance of ( (pressurized) pore water) from Fig. 5a? The data shown in Fig5a are from experiments on simulated calcite gouge carried out under the same effective normal stress and temperature conditions, one lab-dry and the other using a pore fluid pressure of demineralized water of 10 MPa. The inset highlights a part of the slide-hold-slide sequence in the test, demonstrating a marked difference in healing behaviour (i.e., note $\Delta\mu r$). We acknowledge that this was not sufficiently clear in the original ms, and we thank the reviewer for pointing this out. To address this, we now refer to Fig. 5a-inset at the appropriate location in the text, including a note on $\Delta\mu r$ (see line 260-261). Furthermore, we have clarified the text within, and caption to, Figure 4, to better indicate the dry vs. wet experiment.

References NOT cited in the revised ms: Chen, J., Verberne, B. A., and Niemeijer, A. R. Flow-to-friction transition in simulated calcite gouge: Experiments and microphysical modelling. Under review for publication in J. Geophys. Res, preprint available on ESSOAr, 25 April,

Please also note the supplement to this comment:

[Figure]

https://se.copernicus.org/preprints/se-2020-85/se-2020-85-AC2-supplement.pdf